# Oxygen–Ozone Therapy Associated with Alpha Lipoic Acid Plus Palmitoylethanolamide and Myrrh versus Ozone Therapy in the Combined Treatment of Sciatic Pain Due to Herniated Discs: Observational Study on 318 Patients

**DOI:** 10.3390/ijerph19095716

**Published:** 2022-05-07

**Authors:** Matteo Bonetti, Dorina Lauritano, Gian Maria Ottaviani, Alessandro Fontana, Alessio Zambello, Luigi Della Gatta, Mario Muto, Francesco Carinci

**Affiliations:** 1Neuroradiology Service Istituto Clinico Città di Brescia, 25128 Brescia, Italy; dottorbonetti@gmail.com (M.B.); fontanagemini@gmail.com (A.F.); 2Department of Translational Medicine, University of Ferrara, 44121 Ferrara, Italy; crc@unife.it; 3Emergency and Urgency Department, Spedali Civili di Brescia, 25123 Brescia, Italy; gianmaria.ottaviani@gmail.com; 4Anesthesia and Pain Therapy Service, Casa di Cura Borghi, 21020 Brebbia, Italy; a.zambello@gmail.com; 5Neuroradiology Department, Ospedale Cardarelli Napoli, 80131 Napoli, Italy; luigi.dellagatta@hotmail.it (L.D.G.); mutomar2@gmail.com (M.M.)

**Keywords:** lumbar disk herniation, ozone therapy, ozone, alpha-lipoic acid, palmitoylethanolamide, myrrh

## Abstract

Background: The aim of our observational study is to compare the therapeutic efficacy of combined treatment of oxygen–ozone therapy and oral treatment with alpha-lipoic acid (ALA) + palmitoylethanolamide (PEA) and myrrh in patients with peripheral neuropathic pain (sciatica) on radicular disc conflict from disc herniation and the results obtained with oxygen–ozone treatment alone. Methods: We enrolled 318 patients with the neuroradiological diagnosis of disc herniation performed with computed tomography (CT) or magnetic resonance imaging (MRI) and symptoms characterized by low back pain complicated by sciatica, which we divided into two groups. Group A was composed of 165 patients who were treated only with oxygen–ozone therapy with CT-guided intraforaminal technique, while the remaining 153 (Group B) have undergone combined oral treatment with ALA + PEA and myrrh. Follow-up visits for the evaluation of the clinical outcome of the treatment were conducted after 60 ± 8 days using a modified version of McNab’s method. Results: At the clinical check-up, 126/165 patients included in Group A had a complete remission of pain (76.4%), while in Group B, 119/153 (77.8%) had a complete remission of pain. Conclusion: The results highlight how the treatment associated with ozone therapy and oral administration of alpha-lipoic acid + palmitoylethanolamide and myrrh is preferred over the simple treatment with only ozone in such patients in the phase of greatest acuity of the disease, where the pain appears to be better controlled.

## 1. Introduction

Oxygen–ozone therapy for the treatment of herniated discs was introduced in 1985. Over the years, numerous clinical cases have been presented in the literature that has reported positive results ranging from 75% up to almost 90% in the treatment of low back pain, complicated or not by sciatica from disc-radicular impingement, determined by the presence of a herniated intervertebral disc [1,2,3,4,5,6,7,8,9,10,11,12,13,14,15,16,17,18,19,20,21,22].

Low back pain and lumbosciatica are highly disabling pathologies, more and more widespread in every social category, which are manifesting themselves at an increasingly precocious age. They arise acutely, following unusual efforts or movements, or slowly, often with progressive aggravation.

Lumbago and sciatica are painful conditions that are estimated to affect 5 in every 10,000 Western adults. They can be supported by numerous vertebral pathologies, often concomitant: diseases of the disc, of the facet joints, spondylolysis (with or without listesis), somatic and interapophyseal arthrosis, stenosis of the vertebral canal, radicular and synovial cysts, meningiomas, primary or metastatic neoplastic pathology, etc. [6,23].

To choose the best therapy, in cases of low back pain and/or sciatica pain, a precise diagnosis conducted after a careful objective examination and supported by suitable instrumental examinations is therefore essential. In addition to standard radiograms of the spine, in these patients, computed tomography (CT) and/or magnetic resonance imaging (MRI) are therefore of primary importance.

In our study, we exclusively recruited patients with low back pain complicated or not by sciatica pain as by disc-radicular impingement from soft disc hernia (non-calcific).

Among the many causes of low back/lumbosciatalgia, the herniated intervertebral disc is the most common cause of sciatica, affecting 95% of patients with this disorder. The protruding annular fibers of the disc can compress the radicular nerve, causing pain. Oxidative stress potentiates peripheral neuropathy of the lower limbs, reducing neuronal function and local blood flow, resulting in a decrease in the supply of nutrients to the supporting cells.

Considering as a first therapeutic approach, the targeted treatment with oxygen–ozone with intraforaminal CT-guided technique [6], exploiting the anti-inflammatory and analgesic mechanisms of action associated with the possibility of triggering the dehydration mechanism of herniated discs [19,24,25] and the possibility of associating alpha-lipoic acid (ALA) + palmitoylethanolamide (PEA) and myrrh orally, we decided to combine the two therapies considering the known analgesic potential of ALA, which, as it is well known, is a natural sulfur compound produced in small concentrations by all cells. ALA is a key compound in some mitochondrial enzyme complexes (pyruvate dehydrogenase and ketogluterate dehydrogenase), which play a central role in oxidative metabolism. ALA can reduce oxidative stress, preventing damage from oxygen-free radicals [26,27,28,29,30,31,32,33,34,35]. Unlike other antioxidants that have full functionality in aqueous or fatty tissues, ALA exerts its antioxidant function in both water and fats. This property gives thioctic acid a broad spectrum of antioxidant action.

MyrLiq^®^ myrrh is a dry extract of Commiphora myrrha gum resins with a high content of bioactive furanodienes, obtained through a patented extraction process, which allows preserving all the properties of the original raw material. PEA is, on the other hand, a natural compound rich in fatty acids, which, in the body, acts as a biological modulator, favoring the physiological tissue response [36,37,38,39].

ALA/PEA/Myhrr is also effective in the management of complex regional pain syndrome, and as anti-inflammatory, it has been hypothesized that ALA/PEA/M could act on cytokine storm in COVID-19 patients [40,41].

In particular, the use of fast–slow tablets represent a food supplement that supports the normal function of the nervous system. They are also useful in cases of lumbosciatica, carpal or tarsal tunnel syndrome, peripheral neuropathies, alcoholic neuropathies, diabetic neuropathies, and in all those cases in which supplementation of alpha-lipoic acid (thyoptic acid), PEA, and myrrh (MyrLiq^®^) is required.

Considering the above, in this observational study, we wanted to compare the clinical results obtained in the treatment of 318 patients selected for the presence of lumbago and/or sciatica pain, 165 of which were treated only with oxygen–ozone therapy with CT-guided intraforaminal technique, while for remaining 153, we associated the oxygen–ozone therapy treatment with ALA + PEA and oral myrrh.

## 2. Materials and Methods

In this observational study, we re-evaluated the results of the treatment, carried out by CT-guided targeted injection of a gaseous mixture of oxygen–ozone, on 318 patients divided into two groups, treated in the period from January 2019 to December 2021. in Group A, which consisted of 165 patients, treatment was performed exclusively with ozone-depleting pain relief therapy under CT scan with intraforaminal technique. In Group B (153 patients), oxygen–ozone treatment was combined with 800 mg/day of ALA + 600 mg/day of PEA + 200 mg/day of myrrh orally.

We treated a total of 171 males (88 in group A and 83 in group B) and 147 females (79 in group A and 68 in group B) aged between 24 and 66 years (median age: 47.2) with low back pain and/or sciatica. In our study, we exclusively recruited patients with pain as by disc-radicular impingement from soft disc hernia (non-calcific). Written informed consent was obtained from the patients to publish this paper.

On enrollment, the name, date of birth, date of enrollment, date of treatment, and clinical details were recorded for each patient listing the type of pain, irradiation, paranesthesia, Lasègue’s sign, degree of sensitivity, lower limb reflexes, plantar extension of the foot, and dorsal extension of the big toe. Before enrollment, all patients were subjected to CT or MRI scans.

Patients had acute or chronic low back pain and sciatica, which was unilateral or irradiating along with the innervation territories of L3 (19 patients), L4 (159 patients), L5 (105 patients), and S1 (35 patients). Patients with bilateral lower back and sciatic nerve pain and those with electromyographic features of neurogenic injury and/or denervation were excluded.

The injection site was disinfected, and local anesthesia was applied using an ethyl chloride spray in all patients. Infiltrations were done by specialist neuroradiologists at the Neuroradiology Service of Istituto Clinico Città di Brescia (Brescia), Cardarelli Hospital Naples, and Casa di Cura Brebbia (Varese).

## 3. Infiltration Technique

The puncture site was identified by CT scan and marked on the patient’s skin. The distance from this point to the foramen was subsequently measured.

A 22 G Terumo needle (a 9 cm needle was typically used, but longer needles were occasionally adopted depending on the size of the patient) was positioned 2–3 mm from the foraminal region, close to the ganglion of the affected nerve root (Figure 1a). A CT scan was then repeated to check correct needle placement (Figure 1b).

The O_2_-O_3_ was infiltrated by injecting 3 ml of the gas mixture at 25 µg/mL close to the neural foramen, then retracting the needle a few millimeters and injecting another 5 ml of the mixture to involve the facet joint region. CT scans were then used to check the correct distribution of the gas mixture in the foramen and facet joint (Figure 1c).

All treatments were carried out using equipment with photometric detection of the ozone concentration in the gas mixture (the device automatically corrects the deviation of concentration that exists when the syringe withdrawal is carried out) with constant pressure during ozone intake operation. There was total non-toxicity, as the titanium, Teflon, glass, and silicon, which are inert to ozone, were in contact with the gas (Maxi Ozon Active International produced by Medica srl or Alnitec Device, both CE mark class 2A). All patients included in the study underwent three treatments with oxygen–ozone therapy over 30 days with distance of 9 ± 2 days between the first and second therapeutic session and 18 ± 2 between the second and third.

In the 153 patients who were combined with oral administration of 800 mg/day of ALA + 600 mg/day of PEA + 200 mg of myrrh, patients took 2 tablets/day on an empty stomach 30 minutes before breakfast and before dinner (morning and evening). The two assumptions were separated by about ten hours for 20 days of total therapy.

The content of each single tablet provides 800 mg of alpha-lipoic acid, 600 mg of PEA, and 200 mg of myrrh e.s. Patients underwent clinical evaluation after 9 ± 2 days from the first treatment, after 18 ± 2 days from the second treatment, and at the end of the study expected 60 ± 8 days from the start of therapy.

Moreover, the clinical outcome was assessed in all patients; follow-up visits occurred using a modified version of McNab’s method [42,43]:Excellent: resolution of pain and return to normal activity carried out prior to pain onset.Good or satisfactory: more than 50% reduction of pain.Poor: partial reduction of pain below 70%.

## 4. Statistical Analysis

To quantify the time-related change in the proportion of subjects with a poor, good and excellent outcome between the O_2_-O_3_ + ALA (ALA) and the O_2_-O_3_ (CTR) groups, we used an ordinal logistic regression model (Rabe-Hesketh S. Multi-level and longitudinal modeling using Stata. Volume II: Categorical Responses, Counts, and Survival. College Station, TX: Stata Press; 2021). Such model employed the outcome (ordinal: poor, good, and excellent) as response variable and treatment (categorical: 1 =ALA; 0 = CTR), occasion (categorical: 0 = visit I; 1 = visit II; 2 = 60 days), and a treatment X occasion (categorical X categorical) interaction as predictors. The model employed cluster confidence intervals to consider repeated measures. Using this model, the difference in the proportion of patients with a poor, good, and excellent outcome between the ALA and CTR groups on the different occasions was calculated using a within-time between-group contrast with Bonferroni correction for 9 groups (1 difference X 3 outcome levels X 3 occasions). Statistical analysis was performed using Stata 17.0 (Stata Corporation, College Station, TX, USA).

The results of the statistical analysis are reported in Figure 2, which plots the time-related changes in the proportion of patients with a poor, good, and excellent outcome in the CTR and ALA groups, and Figure 3, which plots the between-group (ALA-CTR) difference in the proportion of patients with a poor, good, and excellent outcome during the study.

## 5. Results

At the clinical check-up conducted 60 days ± 8 from the end of the treatment using a modified version of McNab’s method, 116/165 patients in Group A had a complete remission of pain (70.3%) while 21 (12.7%) and 28 (17.0%) had no benefit from the treatment, reporting a partial remission of painful symptoms, while in Group B 119/153 (77.8%) had a complete remission of pain, 13 (8.5%) considered the outcome of the treatment sufficient and 21 (13.7%) insufficient. The patients were also clinically evaluated after the first guided CT infiltration with oxygen–ozone at 9 ± 2 days, and in the patients of Group A, 83/165 (50.3%) reported an excellent result after the treatment, while 37 (22.4%) patients reported a satisfactory and 45 (27.3%) poor clinical result; in Group B, 92/153 patients (60.1%) reported an excellent clinical result, 28 (18.3%) relatively good, and 33 (21.6%) insufficient. In the subsequent intermediate check-up 18 ± 2 days after the second oxygen–ozone therapy session, 94/165 patients in Group A reported an excellent result (57%), while 32 (19.4%) were relatively satisfied, and 39 (23.6%) did not report any improvement; in Group B, the patients who reported an excellent/good clinical result were 106/153 equal to 69.3% (Table 1 and Table 2).

## 6. Discussion

In this observational study, we found that treatment with O_2_-O_3_ + ALA as compared to O_2_-O_3_ was associated with a higher number of excellent results on occasion II (+9%, corresponding to −7% of poor results and −2% of good results, *p* = NS for all), occasion III (+15%, corresponding to −10% of poor cases and−5% of good cases, *p* < 0.05 for all), and 60 days (+7%, corresponding to −4% of poor cases and −3% of good cases, *p* = NS). These promising data suggest the utility of performing a randomized controlled trial to ultimately ascertain the efficacy of ALA as an adjunct to O_2_-O_3_ therapy for the treatment of sciatic pain due to herniated disks.

In recent years, several studies have demonstrated the utility of oxygen–ozone therapy in the treatment of herniated discs [1,2,3,4,5,6,7,8,9,10,11,12,13,14,15,16,17,18,19,20,21,22] as well as the known analgesic potentialities of ALA, while myrrh and the PEA, on the other hand, act as biological modulators, favoring the physiological tissue response [26,27,28,29,30,31,32,33,34,35].

We believe, in this regard, that the administration of intraforaminal ozone CT-guided mode with the associated proposal guarantees control of the needle tract [3,4,5,6].

The possibility of curative oxygen–ozone is, in this regard, high because of the improvement of local circulation with eutrophication in the proximity of the nerve root, compressed and suffering both muscle spasms. It can normalize the level of cytokines and prostaglandins with anti-inflammatory and pain relievers. It increases the production of superoxide dismutase (SOD) with the minimization of oxidizing reagents (reactive oxygen species) [9,10,14,19,24,25]. Finally, the proximity to the herniated material causes accelerated dehydration or destruction of the non-vascularized tissue that justify the good result.

The rapid resolution of pain with no complications, the ease of performing the method and complete control of infiltration by CT allow today to propose intraforaminal CT-guided oxygen–ozone therapy technique as a viable alternative to surgical treatment of disc herniation if the latter is not considered essential and therefore a method of choice between conservative therapies.

## 7. Conclusions

According to the results shown in our sample of 318 patients, both in the intermediate checks at 9 ± 2 days from the first treatment and at 18 ± 2 days from the second treatment, the therapeutic results differ slightly with regard to the patients included in Group A and Group B. 

However, it is possible to see how in the patients included in Group B corresponding to the combined treatment, positive results are assessed slightly more, and certainly, this result can be traced back to the combined action of ALA + PEA + myrrh, which consists of further and effectively relieving the neuropathic symptoms.

Therefore, based on the results, we conclude that the association of a minimally invasive therapy such as oxygen–ozone therapy under CT scan with intraforaminal technique and the oral administration of 800 mg/day of ALA + 600 mg/day of PEA + 200 mg of myrrh to subjects afflicted by low back pain complicated by sciatica can be considered an excellent therapeutic solution, improving the good final clinical result with better control of symptoms, particularly in the first phase of the disease, with the final aim of offering a valid conservative alternative to a possible surgical solution. 

However, we believe that the present study should be extended to an even larger sample of patients.

## Figures and Tables

**Figure 1 ijerph-19-05716-f001:**
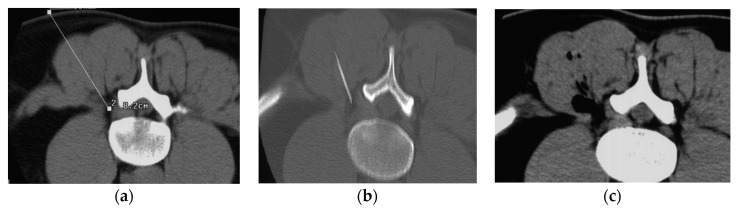
(**a**) Preliminary computed tomography (CT) measurements. The puncture site was identified by CT scan and marked on the patient’s skin. (**b**) Correct positioning of the needle. (**c**) Distribution of the gas mixture.

**Figure 2 ijerph-19-05716-f002:**
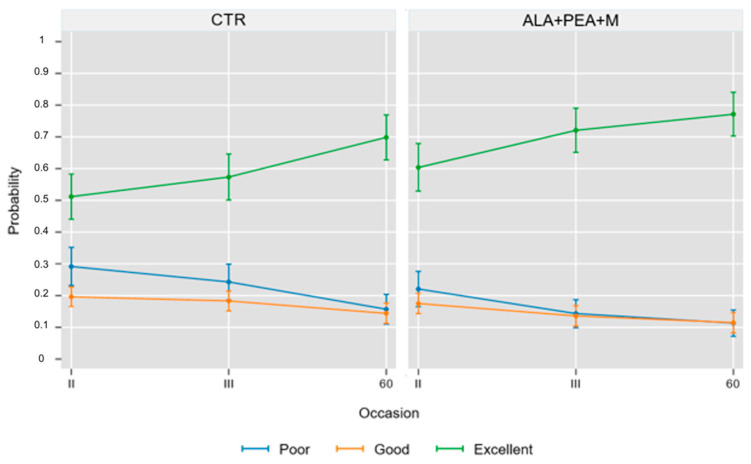
Time-related changes in the proportion of patients with a poor, good, and excellent outcome in the control O_2_-O_3_ (CTR) and O_2_-O_3_ + ALA (ALA) groups. Values are proportions and 95% cluster confidence intervals from ordinal logistic regression (see statistical analysis for details).

**Figure 3 ijerph-19-05716-f003:**
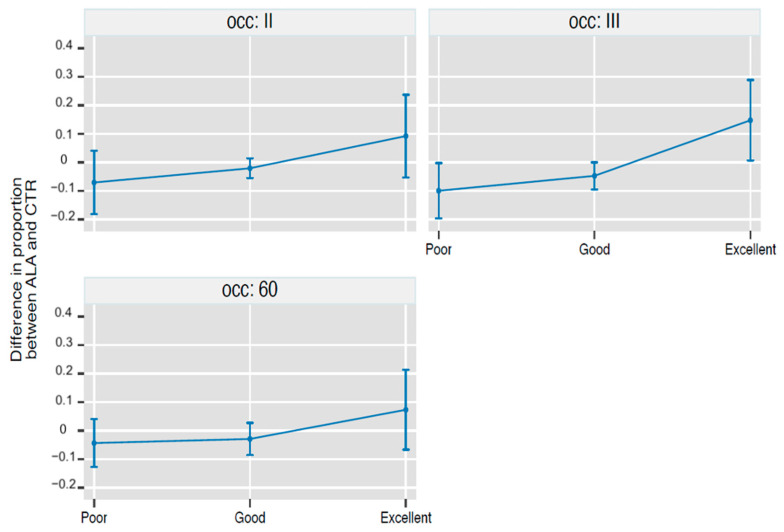
Between-group (control O_2_-O_3_ group and O_2_-O_3_ + ALA group) difference in the proportion of patients with a poor, good, and excellent outcome during the study after 2nd treatment (OCC: II), after 3rd treatment (OCC: III), and at the clinical check-up conducted 60 days ± 8 from the end of the treatment (OCC: 60). Values are proportions and 95% cluster confidence intervals from ordinal logistic regression with Bonferroni correction for 9 contrasts. Points whose 95% CI do not cross the 0 line, i.e., all points at occasion III, are significant at *p*-level < 0.05 (see statistical analysis for details).

**Table 1 ijerph-19-05716-t001:** Outcomes of group A (165 patients) after treatment (ozone-depleting pain relief therapy under CT scan with intraforaminal technique).

Group A	O_2_-O_3_ Treatment		
Outcome	Excellent	Good	Poor
2nd treatment (9 ± 2 days)	83 (50.3%)	37 (22.4%)	45 (27.3%)
3rd treatment (18 ± 2 days)	94 (57%)	32 (19.4%)	39 (23.6%)
60 ± 8 days	116 (70.3%)	21 (12.7%)	28 (17.0%)

**Table 2 ijerph-19-05716-t002:** Outcomes of group B (153 patients) after treatment (oxygen–ozone treatment was combined with 800 mg/day of ALA + 600 mg/day of PEA + 200 mg/day of myrrh orally).

Group B	O_2_-O_3_ Treatment + ALA, PEA, and Myrrh		
Outcome	Excellent	Good	Poor
2nd treatment (9 ± 2 days)	92 (60.1%)	28 (18.3%)	33 (21.6%)
3rd treatment (18 ± 2 days)	106 (69.3%)	22 (14.4%)	25 (16.3%)
60 ± 8 days	119 (77.8%)	13 (8.5%)	21 (13.7%)

## Data Availability

Not applicable.

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
