# Peer review of "Oxygen–Ozone Therapy Associated with Alpha Lipoic Acid Plus Palmitoylethanolamide and Myrrh versus Ozone Therapy in the Combined Treatment of Sciatic Pain Due to Herniated Discs: Observational Study on 318 Patients"

_ijerph, 2022, doi:10.3390/ijerph19095716_

Round 1
Reviewer 1 Report
I want to congratulate the authors for the article entitled: "Oxygen-ozone therapy associated with alpha lipoic acid plus palmitoylethanolamide and myrrh versus ozone therapy in the combined treatment of sciatic pain due to herniated discs. Observational study on 318 patients".
Observational studies are important since they give clues from clinical practice in real world setting.
However, in order to accept the article, I have a few concerns I would like the authors to answer:
- I consider that in page 2, lines 50-51 and 52-55 should be treated as a single paragraph.
- In 7th paragraph (page 2, line 69), authors comment the technique of ozone therapy, and the properties of ALA, PEA and Myhrr, but no reference on ozone properties are described. I suggest to state ozone properties such as a) anti inflammatory, b) anti oxidant, c) favoring nucleolysis and discolysis. Since the effectivennes of ozone in nucleolysis is of 75-90% effectivennes, please reference or comment in Introduction. You could cite: a) Bocci, V. (2005). Ozone A new medical drug. b) Rimeika, G., Saba, L., Arthimulam, G., Della Gatta, L., Davidovic, K., Bonetti, M., ... & Muto, M. (2021). Metanalysis on the effectiveness of low back pain treatment with oxygen-ozone mixture: Comparison between image-guided and non-image-guided injection techniques. European Journal of Radiology Open, 8, 100389. c) de Sire, A., Agostini, F., Lippi, L., Mangone, M., Marchese, S., Cisari, C., ... & Invernizzi, M. (2021). Oxygen–ozone therapy in the rehabilitation field: State of the art on mechanisms of action, safety and effectiveness in patients with musculoskeletal disorders. Biomolecules, 11(3), 356.
- In 9th paragraph in page 2, lines 85, you could add that ALA/PEA/Myhrr is also effective in the management of Complex regional pain syndrome and as anti-inflammatory it has been hypothesized that ALA/PEA/M could act on cytokine storm in COVID-19 patients. You could reference: a) Fernandez-Cuadros, M. E., Albaladejo-Florin, M. J., Martin-Martin, L. M., Alava-Rabasa, S., & Pérez-Moro, O. S. (2020). Effectiveness of Physical Therapy Plus TIOBEC®(α-Lipoic Acid Vitamin B, C, and E) on Complex Regional Pain Syndrome (CRPS): A Small Case Series/Pilot Study. Middle East Journal of Rehabilitation and Health Studies, 7(2). b) Fernández-Cuadros, M. E., Albaladejo-Florín, M. J., Álava-Rabasa, S., Casique-Bocanegra, L. O., López-Muñoz, M. J., Rodríguez-de-Cía, J., & Pérez-Moro, O. S. (2020). Una combinación nutracéutica de ácido alfa-lipoico, palmitoiletanolamida y mirra (TIOBEC-DOL) podría ser eficaz en la prevención de COVID-19.
- In Materials and Methods, page 4, line 153, I believe you could use the brand name of the tablets used, since the formulation is unique. However, I leave that consideration on Editor's Decision and/or Journal's policies.
- In Figure 2, define the acronyms CTR (control) and ALA + PEA + M (alpha lypoic acid+ palmitoylethanolamide + myhrr) in legend.
- In figure 3, please define acronym OCC I, OCC II and OCC 60. Also define ALA and CTR.
- Could you reference the MacNab method used as outcome measure in your manuscript?
- Is threre any limitation observed in your study? Sample size?
Once these few issues are answered I could suggest your article for publication, beacuse of the innovative results observed in the management of sciatic pain due to herniated discs.
Author Response
Ferrara, 29th April 2022
Dear editor:
Many thanks for the insightful comments and suggestions of the referees. We have made corresponding revisions according to their advice. Words in red are the changes we have made in the text.
- I consider that on page 2, lines 50-51 and 52-55 should be treated as a single paragraph. We treated lines 50-51 and 52-55 as a single paragraph.
- In the 7th paragraph (page 2, line 69), the authors comment on the technique of ozone therapy, and the properties of ALA, PEA, and Myhrr, but no reference to ozone properties is described. I suggest stating ozone properties such as a) anti-inflammatory, b) antioxidant, and c) favoring nucleolysis and discolysis. Since the effectiveness of ozone in nucleolysis is of 75-90% effectiveness, please reference or comment on the Introduction. You could cite: a) Bocci, V. (2005). Ozone is A new medical drug. b) Rimeika, G., Saba, L., Arthimulam, G., Della Gatta, L., Davidovic, K., Bonetti, M., ... & Muto, M. (2021). Metanalysis on the effectiveness of low back pain treatment with an oxygen-ozone mixture: Comparison between image-guided and non-image-guided injection techniques. European Journal of Radiology Open, 8, 100389. c) de Sire, A., Agostini, F., Lippi, L., Mangone, M., Marchese, S., Cisari, C., ... & Invernizzi, M. (2021). Oxygen–ozone therapy in the rehabilitation field: State of the art on mechanisms of action, safety, and effectiveness in patients with musculoskeletal disorders. Biomolecules, 11(3), 356.
We added comments and references on ozone properties in the 7th paragraph.
- In 9th paragraph in page 2, lines 85, you could add that ALA/PEA/Myhrr is also effective in the management of Complex regional pain syndrome and as anti-inflammatory it has been hypothesized that ALA/PEA/M could act on cytokine storm in COVID-19 patients. You could reference: a) Fernandez-Cuadros, M. E., Albaladejo-Florin, M. J., Martin-Martin, L. M., Alava-Rabasa, S., & Pérez-Moro, O. S. (2020). Effectiveness of Physical Therapy Plus TIOBEC®(α-Lipoic Acid Vitamin B, C, and E) on Complex Regional Pain Syndrome (CRPS): A Small Case Series/Pilot Study. Middle East Journal of Rehabilitation and Health Studies, 7(2). b) Fernández-Cuadros, M. E., Albaladejo-Florín, M. J., Álava-Rabasa, S., Casique-Bocanegra, L. O., López-Muñoz, M. J., Rodríguez-de-Cía, J., & Pérez-Moro, O. S. (2020). Una combinación nutracéutica de ácido alfa-lipoico, palmitoiletanolamida y mirra (TIOBEC-DOL) podría ser eficaz en la prevención de COVID-19.
We added the following sentence in the 9th paragraph: “ALA/PEA/Myhrr is also effective in the management of Complex regional pain syndrome and as anti-inflammatory it has been hypothesized that ALA/PEA/M could act on cytokine storm in COVID-19 patients (42,43)”.
- In Figure 2, define the acronyms CTR (control) and ALA + PEA + M (alpha lypoic acid+ palmitoylethanolamide + myhrr) in legend.
We defined the acronyms CTR and ALA+PEA+M in the legend of Figure 2.
- In figure 3, please define the acronyms OCC I, OCC II, and OCC 60. Also, define ALA and CTR.
We defined the acronyms OCC I, OCC II, OCC: 60, and ALA-CTR in Figure 3.
- Could you reference the MacNab method used as an outcome measure in your manuscript?
We added references to the McNab method used as an outcome measure.
- Is there any limitation observed in your study? Sample size?
We explained in the “Conclusion” section that our study should be extended to a larger sample of patients.
Thank you for receiving our manuscript and considering it for publication.
We appreciate your time and look forward to your response.
Yours sincerely,
Prof. Dorina Lauritano

Reviewer 2 Report
The article entitled "The article entitled "A" is of interest because it proposes a complementarity between ozone therapy and a complementary treatment through supplementation. The foregoing in the context of a very frequent painful pathology in the population." is of interest because it proposes a complementarity between ozone therapy and a complementary treatment through supplementation. The foregoing in the context of a very frequent painful pathology in the population.
However, a detailed review of it is necessary before being published. Summary, line 31 delete from ... check-up... to ... the end". The study period of 60 plus minus 8 days is repeated.
Introduction and the entire manuscrip: the bibliographical references do not follow an order (example, line 45, 55, 77, etc.). Lines 61 and 62 correspond to materials and methods. Abbreviations are not respected, for example PEA, it is abbreviated in lines 71, 82, 89 and 95. Materials and methods: ethical approval of the study is lacking. Median age, line 106, should be the median. Line 116 and throughout the text: avoid single paragraph sentences (examples: lines 125,159 to 163, 234).
International System of Units, line 131 and 132, must be mL and not cc. Statistical analysis, lines 158 to 161, appears in results, lines 189 to 206. Tables 1 and 2, should be merged, legend is missing. Table 1 and 2, statistical analysis and figures, the number of observations (medical visits, appears indistinctly as: treat, visit or occasion), homogenize. Figure 3, incomprehensible. Discussion: There are no bibliographic citations (re-write). Line 240, abbreviation ROS, inappropriate. Conclusions: the need for larger clinical studies should be suggested.
Author Response
Ferrara, 29th April 2022
Dear editor:
Many thanks for the insightful comments and suggestions of the referees. We have made corresponding revisions according to their advice. Words in red are the changes we have made in the text.
- Summary, line 31 delete from ... check-up... to ... the end". The study period of 60 plus minus 8 days is repeated.
We deleted the repeated sentence in the Abstract.
- Introduction and the entire manuscript: the bibliographical references do not follow an order (for example, lines 45, 55, 77, etc.).
We reorganized the bibliographical references.
- Lines 61 and 62 correspond to materials and methods.
We reported the sentence also in the “Materials and Methods” section.
- Abbreviations are not respected, for example, PEA is abbreviated in lines 71, 82, 89, and 95.
We uniformed the abbreviations.
- The median age, line 106, should be the median.
We substituted the word “mean” with the word “median age”.
- Line 116 and throughout the text: avoid single paragraph sentences (examples: lines 125,159 to 163, 234).
We reorganized the text, avoiding single paragraph sentences.
- International System of Units, lines 131 and 132, must be mL and not cc. We corrected the System of Units.
- Statistical analysis lines 158 to 161, appear in the results, lines 189 to 206. We reported the “Statistical analysis” in the “Materials and Methods” section.
- Tables 1 and 2, should be merged, the legend is missing. Tables 1 and 2, statistical analysis and figures, the number of observations (medical visits, appears indistinctly as treat, visit or occasion), homogenize.
We reorganized and homogenized Tables and Figures.
- Figure 3, incomprehensible. We modified Figures 2 and 3.
- Discussion: There are no bibliographic citations (re-write). We added bibliographic citations in the “Discussion” section.
- Line 240, abbreviation ROS, inappropriate. We corrected the abbreviation “ROS”.
- Conclusions: the need for larger clinical studies should be suggested. We added this suggestion in the “Conclusion” section.
Thank you for receiving our manuscript and considering it for publication.
We appreciate your time and look forward to your response.
Yours sincerely,
Prof. Dorina Lauritano

Round 2
Reviewer 2 Report
Minor changes needed
Lines 159, 160, 268 and Thought the text, leave a space between the value and its unit. e.g. 200 mg instead of 200mg.
Line 189, Figure 3 and thought the text. Clearly define the times of the consultations and their nomenclatures, in a homogeneous way. e.g. line 189, it should say 9 +/- 2 days instead of 7/10 days and 18 +/- 2 days instead of 15/20 days. In materials and methods, a unique nomenclature must be defined for the consultation times, eg. it seems that OCC II corresponds to 9 +/- 2 d, and OCC III to 18 +/- 2 d, while OCC60 should correspond to 60 +/- 8 d. Lines 220 and 222. Capital letters in Excelent and Goog. Check all the punctuation marks of the numbers in the table. To separate decimals use only the point.
Author Response
Ferrara, 04th May 2022
Dear editor:
Many thanks for the insightful comments and suggestions of the referees. We have
made corresponding revisions according to their advice. Words in red are the changes we have made in the text.
- Lines 159, 160, 268 and Thought the text, leave a space between the value and its unit. e.g. 200 mg instead of 200mg.
We corrected the values and the units following your suggestions.
- Line 189, Figure 3 and thought the text. Clearly define the times of the consultations and their nomenclatures, in a homogeneous way. e.g. line 189, it should say 9 +/- 2 days instead of 7/10 days and 18 +/- 2 days instead of 15/20 days. In materials and methods, a unique nomenclature must be defined for the consultation times, eg. it seems that OCC II corresponds to 9 +/- 2 d, and OCC III to 18 +/- 2 d, while OCC60 should correspond to 60 +/- 8 d.
We homogenized the times and their nomenclatures.
- Lines 220 and 222. Capital letters in Excellent and Good.
We corrected these words.
- Check all the punctuation marks of the numbers in the table. To separate decimals use only the point.
We corrected numbers using only points
Thank you for receiving our manuscript and considering it for publication.
We appreciate your time and look forward to your response.
Yours sincerely,
Dorina Lauritano
